# Macrophages in Skin Wounds: Functions and Therapeutic Potential

**DOI:** 10.3390/biom12111659

**Published:** 2022-11-08

**Authors:** Seen Ling Sim, Snehlata Kumari, Simranpreet Kaur, Kiarash Khosrotehrani

**Affiliations:** 1The University of Queensland Diamantina Institute, Faculty of Medicine, Translational Research Institute, The University of Queensland, 37 Kent Street, Woolloongabba, QLD 4102, Australia; 2Mater Research Institute-UQ, Translational Research Institute, Brisbane, QLD 4102, Australia

**Keywords:** wound healing, macrophage, chronic wounds, inflammation, wound regeneration, NF-κB, cytokine signalling

## Abstract

Macrophages regulate cutaneous wound healing by immune surveillance, tissue repair and remodelling. The depletion of dermal macrophages during the early and middle stages of wound healing has a detrimental impact on wound closure, characterised by reduced vessel density, fibroblast and myofibroblast proliferation, delayed re-epithelization and abated post-healing fibrosis and scar formation. However, in some animal species, oral mucosa and foetal life, cutaneous wounds can heal normally and remain scarless without any involvement of macrophages. These paradoxical observations have created much controversy on macrophages’ indispensable role in skin wound healing. Advanced knowledge gained by characterising macrophage subsets, their plasticity in switching phenotypes and molecular drivers provides new insights into their functional importance during cutaneous wound healing. In this review, we highlight the recent findings on skin macrophage subsets, their functional role in adult cutaneous wound healing and the potential benefits of targeting them for therapeutic use.

## 1. Introduction

The skin is the largest organ in the body and acts as a physical barrier against external mechanical, biological or chemical stimuli to protect the body’s physiological function and internal organs. It consists of three different layers. The outer protective layer is called the epidermis, which hosts keratinocytes, Langerhans cells (LCs) and melanocytes. The middle layer, the dermis, which is rich in immune cells, the extracellular matrix (ECM), fibroblasts, blood vessels and skin appendages, provides nutrition, support and structure to the skin. Below the dermis lies an adipocyte-rich layer called the hypodermis, which further connects the skin with the connective tissue below, provides growth factors to the dermis and acts as an energy reservoir. The skin barrier can be disrupted by being wounded, and to maintain internal organ homeostasis, there is an evolutionarily conserved process to close wounds in adults. The various cellular components of all three layers synchronously participate in multiple stages of skin wound healing, among which a high abundance of skin macrophages (Mϕ) play a central role [1,2]. Dysregulation in any wound-healing phase, particularly in macrophages, could prominently impair the healing process and lead to chronic wound development [2,3,4]. Therefore, understanding macrophages’ heterogeneity and their molecular crosstalk with other cell types during wound repair is necessary to ameliorate wound-healing outcomes. This review will summarise the role of various macrophage subsets in skin homeostasis and cutaneous wound healing. Notably, we will focus on the molecular crosstalk between macrophages and other cell types, such as fibroblasts, keratinocytes, endothelial cells (ECs) and neutrophils, within the granulation tissue during murine acute cutaneous wound repair. In addition, we have highlighted the microenvironmental changes in murine and human chronic wounds and their consequences on macrophages’ physiological function.

## 2. Macrophage Subsets and Their Plasticity in Healthy Skin

Macrophages are highly heterogenous, plastic cells that can undergo phenotypic switching based on external stimuli. Healthy skin is colonised by different subsets of macrophages at specific anatomical locations. Advanced fate mapping, single-cell transcriptomics and imaging have revealed the ontologically distinct functions of macrophages and their roles in maintaining skin homeostasis [5,6,7,8]. The epidermis harbours LCs, a unique population of mononuclear phagocytes, interspersed between melanocytes and keratinocytes [9,10]. These cells act as the first line of defence and are known to be of dual origin. In mice, LCs are derived from the yolk sac primitive macrophages around E9.5–E10.0 and are replenished from foetal liver monocytes between E13.5 and E16.5 [11]. They seed in the epidermis early in their development and self-renew independently of blood monocytes throughout adulthood [11,12,13,14]. Although LCs’ development resembles that of tissue-resident macrophages, their physiological functions are more similar to those of dendritic cells (DCs), which have migratory properties. Therefore, the functional role of LCs in cutaneous wound healing is not covered in this review.

Dermal macrophages are equally derived from yolk sacs and/or foetal livers, but one subset of dermal macrophages is constantly replenished by bone marrow (BM) monocytes after inflammation/injury [15,16], whereas the other small subset is long-lived and has a self-renewal capacity [8]. Intriguingly, a transcriptomic analysis revealed that monocyte-derived macrophages can undergo epigenetic reprogramming and adopt a similar gene expression profile to the tissue-resident macrophages residing within a specific sub-tissular microenvironment after bone marrow transplantation [17]. These results indicate that perhaps macrophage identity is predominantly shaped by local tissue microenvironments, regardless of the macrophages’ ontogeny, thus further highlighting the diversity, complexity and plasticity of dermal macrophages. In the section below, we have discussed various macrophage subsets in the murine and human dermis, as well as highlighted their functional roles in maintaining skin homeostasis. 

### 2.1. Murine Dermal Tissue-Resident Macrophages

#### 2.1.1. Perineural Macrophages

Perineural macrophages are a recently identified, small subset of prenatally derived (C-X3-C motif) chemokine receptor 1 (CX3CR1)^Hi^ tissue-resident macrophages with a self-renewal capacity [8]. During homeostasis, they patrol the skin’s sensory nerve axons and promote nerve myelin degradation after injury [7,8]. Despite that, CX3CR1^Hi^-resident macrophages do not contribute to the repopulation of perineural macrophages around the newly formed unmyelinated axons in wound lesions [8]. Interestingly, the other dermal-resident macrophage subset, specifically that containing CX3CR1^Lo^ macrophages, is associated with sprouting nerves and acquires a high level of CX3CR1 expression progressively after injury (Figure 1B (i)) (Table 1) [8].

#### 2.1.2. Perivascular Macrophages

Perivascular macrophages are a distinct population of skin macrophages characterised by a close association with the skin’s vasculature. These are mainly (C-C motif) chemokine receptor 2 (CCR2)^Neg^, CX3CR1^Neg/Lo^, CD206^Hi^, CD86^Hi^ and Lyve1^Hi^; they display anti-inflammatory properties, are highly phagocytic and UV-sensitive but are radiation-resistant [6,7]. In healthy skin, perivascular macrophages extend protrusions between endothelial intercellular junctions, gaining access to the vascular lumen and taking up particles in circulation [6]. The selective depletion of perivascular macrophages hinders skin repair by reducing angiogenesis, granulation tissue formation, collagen deposition and re-epithelisation (Table 1) [6]. Similarly, another population of CD4^+^ perivascular macrophages is also localised adjacent to postcapillary venules in the skin [5]. These CD4^+^ perivascular macrophages assist in neutrophil recruitment and extravasation during bacterial infections [5]. However, the lineage-tracing study and single-cell RNA sequencing do not correlate CD4^+^ perivascular macrophages to either CX3CR1^Hi^ or CX3CR1^Neg/Lo^ dermal macrophages and thus further highlight the heterogeneity and complexity of macrophages [6,7,8].

#### 2.1.3. Perifollicular Macrophages

A subset of dermal-resident macrophages is associated with keratin (Krt)15^+^Lgr5^+^ hair follicle (HF) progenitors, including the bulge and hair germ [21,22]. The slow-cycling bulge and fast-cycling hair germ are responsible for the hair-cycling process and contribute to epidermal regeneration after cutaneous injuries [23]. Throughout one’s life, the HF undergoes cycles of telogen (resting phase), anagen (growing phase) and catagen (regression phase). The hair cycle is regulated by stimulatory Wnt/β-catenin signalling and inhibitory bone morphogenetic protein (BMP) factors present in the perifollicular niches [24]. Early observations have shown that macrophage numbers reduce concomitantly with the onset of anagen, suggesting a potential role of macrophages in regulating the hair cycle [22]. It was later revealed that perifollicular macrophages undergo apoptosis during late telogen, releasing Wnt7b and Wnt10a into the perifollicular niche [22,25]. The Wnt ligands then activate Wnt/βcatenin signalling in the Lgr5^+^ hair germ and induce hair growth [22]. The importance of macrophages in the HF’s entry into anagen was further supported by macrophage depletion studies in which the removal of macrophages shortened the telogen phase [22]. Similarly, a recent study identified a triggering receptor expressed on myeloid cells 2 (TREM2)^+^ macrophages predominating in the early telogen phase [21]. The ablation of TREM2^+^ macrophages induces hair growth and the proliferation of HF progenitors, suggesting these macrophages are spatially, temporally and functionally relevant for HF progenitors’ quiescence during telogen [21].

### 2.2. Human Dermal-Resident Macrophages

It is apparent that different tissue-resident macrophages reside at specific sub-tissular niches within the murine dermis. Although emerging evidence has indicated the existence of CD163^+^ perivascular and CD64^+^ perineural macrophages in the human dermis [6,8], the human homolog of murine dermal-resident macrophages remains ambiguous. Traditional studies have identified human macrophages based on the positive expression of HLA-DR, CD14, CD11b, CD1C, CD68 and colony-stimulating factor 1 receptor (CSF1R). Nevertheless, the characterisation of human dermal-resident macrophages was hindered by a lack of unique markers. Patients devoid of blood monocytes have normal tissue-resident macrophages in their skin biopsy, suggesting that long-lived and self-renewing macrophage subsets also exist in the human dermis [26]. Single-cell sequencing has also identified several subsets/cell states of CD68^+^ macrophages in adult skin [27,28,29]. 

Among all, macrophage receptor with collagenous structure (MARCO)^+^ macrophages were consistently identified in two studies and express high levels of the complement proteins’ (C1QB and C1QC) transcripts and the cell surface receptor CD163 [27,28]. Xue et al. also identified two macrophage subsets in which CCR1^+^ macrophages have an inflammatory phenotype and express high levels of (C-X-C motif) ligand 2 (CXCL2), (C-C motif) ligand 3 (CCL3), CCL4 and CEBOB, and TREM2^+^ macrophages, which are highly enriched in lipid metabolism [28]. Another population of Mac2^+^ macrophages displaying an alternative activation state with upregulated NR4A1, NR4A2 and KLF4 has also been reported in human skin and can be identified by F13A1 [27]. Remarkably, Mac2 macrophages’ transcriptomic profile closely aligns with that of foetal macrophages in embryonic skin [27]. Skin lesions from patients with atopic dermatitis and psoriasis have increased Mac2 macrophages [27]. A gene ontology analysis has revealed that this population is enriched with stress, chemotactic and angiopoietin genes, suggesting Mac2 might help regulate angiogenesis, immune cell recruitment and TGF-β signalling [27]. Receptor–ligand and in situ analyses show that Mac2^+^ cells might interact with venular vascular endothelial cells via CXCL8/ACKR1 signalling [27].

Based on single-cell RNA datasets, there have been similarities in gene expression between the murine and human macrophage subpopulations [7,8,27,28]. However, to what extent these human dermal macrophages functionally resemble murine dermal macrophages is still poorly understood. Similarly, the tissue-specific niches for each of these macrophage subsets remain largely unknown. Therefore, spatial transcriptomic analyses of healthy human skin and lesions from different pathophysiological disorders (e.g., acute and chronic wounds) may prove to be a valuable tool for the advancement of and knowledge about human skin macrophage biology. 

### 2.3. Monocyte-Derived Macrophages

Monocyte-derived macrophages contribute immensely to cutaneous wound healing. Two subsets of monocytes, namely CCR2^+^Ly6C^Hi^ (CD14^+^CD16^Neg^ in humans) and CX3CR1^+^Ly6C^Lo^ (CD14^Neg^CD16^+^ in humans), travel from adult bone marrow to the injury site when chemokines CCL2 and CX3CL1 are released [30,31]. Upon extravasation into interstitial tissue, monocytes can differentiate and polarize into two macrophage subtypes, the classical macrophages with a pro-inflammatory phenotype or the non-classical macrophages with a pro-healing phenotype, based on cues from their microenvironment. The distinctive pro-inflammatory and pro-healing roles that macrophages play in different phases of the murine acute wound-healing process will be discussed in the following sections. 

## 3. Macrophages’ Role in Various Stages of Wound Healing 

Acute skin wound healing is a cascade of well-orchestrated signalling events in various cell types at different stages. This complex process is divided into four sequential and overlapping phases: haemostasis, inflammation, proliferation and remodelling. A dysregulation of molecular signals in the wound microenvironment at any stage of the healing process will affect the rate of wound closure and scarring [32]. Importantly, macrophages undergo specific phenotypic and functional changes to play a critical role in all stages of the healing process. Any dysregulation in the macrophages’ function is associated with delayed wound healing and scarring. Chronic wounds and fibrosis are such pathophysiological conditions, caused by prolonged inflammation due to the accumulation of pro-inflammatory macrophages that were unable to switch to a pro-healing phenotype, also known as anti-inflammatory macrophages [1]. 

### 3.1. Haemostasis Phase (Day 0)

Haemostasis starts seconds after the skin is wounded. Blood vessels rupture during acute injury, triggering vasoconstriction and the activation of platelets and tissue-resident macrophages [33]. To prevent further blood loss and pathogen invasion, platelets aggregate and catalyse a series of coagulation processes in the open wounds [33,34]. Platelets are activated when subendothelial basement membrane proteins, such as fibronectin and collagen, bind to platelet cells’ surface receptors [34]. This process induces platelet aggregation and degranulation, releasing chemical mediators that successively activate more platelets via a positive feedback loop [34]. In addition, thromboxane and serotonin released by platelets help maintain the vasoconstriction by acting on endothelial smooth muscle [35,36]. A series of coagulation factors released by platelets convert inactive prothrombin into activated thrombin [34]. Thrombin catalyses inactive fibrinogen into active fibrin monomers [34]. Together with interstitial collagen, erythrocytes, platelet aggregates and other blood components, a provisional ECM of fibrin fibres is formed by fibrin intermolecular crosslinking which is stabilised by coagulation factor XIIIa [34]. Notably, the provisional fibrin fibres also serve as a scaffold for cell recruitment and direct immune cells’ migration in the subsequent wound-healing phases. 

### 3.2. Macrophages in the Inflammatory Phase (Day 1–3/4)

The inflammatory phase happens soon after haemostasis. Immediately after injury, pattern-recognition receptors (PRRs) present on the tissue-resident macrophages in the dermis are activated by “danger-associated molecular patterns” (DAMPs) released by damaged and necrotic cells, and “pathogen-associated molecular patterns” (PAMPs) on pathogens [37]. In response to the stimuli, activated platelets and tissue-resident macrophages produce pro-inflammatory chemokines (e.g., CXCL-1,5,8; CCL2), cytokines (e.g., tumour necrosis factor (TNF), interferon (IFN)-γ, interleukin (IL)-1β, IL-6, IL-33) and growth factors, which initiate the recruitment and activation of other immune cells, mainly neutrophils and bone-marrow-derived monocytes into the wound, propagating the inflammation to remove foreign particles and damaged cells (Figure 1A (i)) [33,38,39]. Neutrophils and monocytes are recruited either via microhaemorrhage or transendothelial migration into the injury site [39]. Apart from the major cytokines and chemokines, reactive oxygen species (ROS), such as hydrogen peroxide (H_2_O_2_), also promote neutrophil recruitment and macrophage polarisation [40,41]. The reduced H_2_O_2_ levels in murine and zebrafish wound-healing models decrease the neutrophil numbers and TNF-expressing macrophages without affecting macrophage chemotaxis [41]. Importantly, the transcriptional activity of the nuclear factor kappa B (NF-κB), the master regulator of pro-inflammatory genes, is reduced concomitantly with the H_2_O_2_ levels, indicating that H_2_O_2_-induced neutrophil recruitment and pro-inflammatory macrophage polarisation are regulated downstream of the NF-κB signalling [40,41]. 

#### 3.2.1. Macrophage Regulation of Neutrophil Recruitment and Pro-Inflammatory Macrophage Polarisation

Neutrophils are the most abundant cells found in wounds during the early stage of inflammation and their numbers decrease after 4 days post-wounding [32]. Tissue-resident perivascular macrophages regulate the extravasation of neutrophils by regulating the intercellular adhesion molecule 1 (ICAM-1) expression on ECs [5,42]. The expression of adhesion molecules, such as P-selectin, E-selectin, vascular cell adhesion protein 1 (VCAM-1) and ICAM-1, on ECs is necessary for neutrophil attachment to the endothelium lining before it can transmigrate into the interstitial tissue via the endothelial tight junction [43]. Infiltrated neutrophils combat pathogen invasion by releasing ROS, nitrogen oxide (NO), antimicrobial proteins and proteolytic enzymes, and then forming the neutrophil extracellular trap (NET) before phagocytosing the invaders (Figure 1A (ii)) [44,45]. Despite their beneficial roles, neutrophils’ persistence and increased protease activity can induce tissue damage, hence contributing to delayed wound healing.

Similarly, there is a release of CCL2 from various tissue-resident cells into the bloodstream after an insult which triggers the infiltration of CCR2^+^Ly6C^Hi^ monocytes during the early stage of wound healing (Figure 1A (iii)). These CCR2^+^Ly6C^Hi^ monocytes then polarise into pro-inflammatory Mϕ in response to the pro-inflammatory signals released, such as IFN-γ, TNF, DAMPs or PAMPs (e.g., the lipopolysaccharides (LPS) present in the wound microenvironment (Figure 1A (iv))) [46]. Both neutrophils and pro-inflammatory macrophages play vital roles as scavengers, phagocytose cellular debris and necrotic cells during the inflammatory phase (Figure 1A (v)) [32]. They produce anti-microbial peptides, pro-inflammatory cytokines, chemokine and ROS to prevent pathogen colonisation and to further enhance monocyte/macrophage recruitment [33,47].

#### 3.2.2. Macrophage Regulation of Neutrophils and Anti-Inflammatory Macrophage Polarisation

Immediately after the clearance of foreign debris, neutrophils undergo apoptosis and are removed by pro-inflammatory macrophages in a process called efferocytosis [33]. Macrophage–neutrophil physical interactions help to direct neutrophil migration back into circulation (reverse migration) and shorten the traveling time within the wound as another mode of inflammation resolution [48,49]. Macrophage depletion in the wound results in less-effective neutrophil reverse migration and prolonged neutrophil accumulation in the wound [49].

The phagocytosis of neutrophils is important for the wound to progress into an anti-inflammatory microenvironment [50]. Efferocytosis induces a reparative phenotype in macrophages by inducing the production of anti-inflammatory cytokine and growth factor, such as transforming growth factors β (TGF-β), IL-10 and vascular endothelial growth factor (VEGF) (Figure 1B (ii)) [51]. Diabetic ulcers, a common chronic wound worldwide, display high neutrophil numbers, delayed macrophage infiltration, neutrophil clearance, pro-inflammatory macrophage accumulation and an impaired transition to an anti-inflammatory phenotype [52]. These phenomena could directly or indirectly relate to macrophage dysfunction as a high-glucose condition is known to reduce macrophages’ phagocytic capability [53,54]. Consequently, neutrophils may accumulate within diabetic wounds due to impaired macrophage phagocytosis; hence, prohibiting the anti-inflammatory phenotype transition. These cumulative factors eventually contribute to the persistence of inflammation and finally cause tissue damage. 

Although exacerbated inflammation can cause significant tissue damage, the infiltration of the pro-inflammatory monocytes/macrophages in the early phase of wound healing is necessary for the wound-healing progression. *CCR2*-deficient mice, which have reduced levels of circulating CCR2^+^Ly6C^Hi^ monocytes and pro-inflammatory macrophages in their wounds, showed impaired wound closures (Table 1) [18]. The adoptive transfer of CCR2^wild-type^ monocytes to *CCR2*^−/−^ mice or the administration of CCL2, the ligand of CCR2, during the early stage of wound healing in diabetic wounds can rescue delayed wound healing (Table 1 and Table 3) [18,55,56]. Similarly, the conditional abrogation of macrophages in the early phase of wound healing using *LysMCre-DTR* mice also results in reduced vessel density, fewer myofibroblasts, impaired epithelisation and reduced scar formation [2]. Interestingly, the administration of pro-healing macrophages into the early stage of a diabetic wound does not improve wound healing but shows delayed re-epithelialisation accompanied by an increased frequency of vascular leakage, immature granulation, and the persistence of neutrophils [57]. These have further demonstrated that wound healing is a highly complex process and the spatiotemporal existence of different macrophage subsets is required to perform specific roles in each of the wound-healing phases, driving wound-healing progression.

### 3.3. Macrophages in the Wound Proliferation Phase (Day 3/4–7)

Wound healing progresses into the proliferation stage when inflammation dampens. At this stage, the wound microenvironment changes from a pro-inflammatory into a pro-healing environment. This is accompanied by a phenotype switch of pro-inflammatory macrophages into anti-inflammatory macrophages (Ly6C^Lo^CX3CR1^Hi^) [31]. Anti-inflammatory macrophages expressing high levels of CX3CR1 are the most predominant cell type in the granulation tissue during the proliferation phase [31]. These macrophages are highly immunosuppressive, have regenerative properties and secrete multiple angiogenic and growth factors, cytokines and chemokines (e.g., metalloproteinases (MMPs), platelet-derived growth factor (PDGF), resistin-like molecule α (RELM-α), VEGF, IL-8, TGF-β, IL-10, and arginase) which are essential for the recruitment and activation of various cell types. These factors induce the proliferation and differentiation of the ECs, keratinocytes and fibroblasts along with the deposition of the ECM to reconstruct the skin’s integrity [33]. As such, macrophage depletion during the proliferation stage has a direct or indirect influence on wound revascularisation, matrix production and re-epithelisation [58]. For example, mice defective in CX3CR1 have delayed wound healing characterised by reduced macrophage and myofibroblast numbers, granulation tissue, collagen deposition, growth factors and vessel density (Table 1) [19].

To date, the molecular modulators that induce anti-inflammatory macrophage polarisation in cutaneous wounds in vivo are still largely unclear. IL-4, IL-13 and IL-10 are commonly used to induce various subtypes of anti-inflammatory macrophages in vitro. Macrophages stimulated with IL-4 and IL-13 acquire a M2a phenotype and promote type II inflammatory responses. Nevertheless, IL-4 and IL-13 are not identified in wound fluid [59]. IL-4Rα knockout mice have a normal level of anti-inflammatory macrophages in a polyvinyl alcohol (PVA) sponge wound model, suggesting the involvement of other cytokines in anti-inflammatory macrophage polarisation in vivo [59]. On the other hand, the stimulation of macrophages with IL-10 gives rise to M2c macrophages. IL-10 exerts potent anti-inflammatory effects by downregulating pro-inflammatory cytokine productions in different immune cells, including macrophages [60]. Emerging evidence has shown that CCR2^+^Ly6C^Hi^ cells can gradually lose Ly6C expression and give rise to CX3CR1^+^Ly6C^Lo^ monocytes in the periphery [61,62]. The transcription factor Nr4a1 is essential for the conversion of Ly6C^Hi^ to Ly6C^Lo^ monocytes [63]. Similarly, the characterisation of the wound macrophage phenotype in vivo has never adopted an absolute pro-inflammatory or anti-inflammatory phenotype but rather has a continuum spectrum of polarization stages with pro-inflammatory and anti-inflammatory macrophages at the extreme ends. Using the CX3CR1^gfp/+^:CCR2^rfp/+^ mice, Rahmani et al. captured the dynamic of CCR2^+^ and CX3CR1^+^ macrophages at various stages of the cutaneous-wound-healing process [31]. It was found that most wound macrophages express both CCR2 and different levels of CX3CR1, namely CX3CR1^Lo^, CX3CR1^Med^ and CX3CR1^Hi^ [31]. The percentage of CX3CR1^Lo^ and CX3CR1^Med^ macrophages (the majority were Ly6C^Hi^) were highest during the early stage of wound healing while CX3CR1^Hi^ (mostly Ly6C^Lo^) predominated in the later stage [31]. There was a reduced number of total CD11b^+^ and CX3CR1^Hi^ macrophages in the wounds of CX3CR1-deficient mice [31]. Nevertheless, CX3CR1 is required for monocytes’ survival and the reduced CX3CR1^Hi^ macrophages might be due to increased cell death [64]. Therefore, it is still unclear whether the anti-inflammatory CX3CR1^Hi^Ly6C^Lo^ macrophages are recruited as a new population or whether they emanate from a switch in inflammatory macrophages’ phenotypes during the proliferation phase of wound healing. Notably, the ratio of pro-inflammatory and anti-inflammatory macrophages is the main factor that shapes the local wound microenvironment, with a higher proportion of pro-inflammatory macrophages during the inflammatory phase and a higher number of anti-inflammatory macrophages during the proliferation phase [65]. This reflects the phenotypic plasticity and complexity of the monocyte/macrophages in response to external stimuli and the importance of their phenotype switching for wound healing to progress.

#### 3.3.1. Macrophage Regulation of Wound Angiogenesis

Proper revascularisation in the granulation tissue is required to support the oxygen and nutrient demands of the actively proliferating cells, as well as the removal of toxic waste and carbon dioxide from the tissues [2,19,38]. The adequate re-oxygenation of the granulation is necessary to permit complete wound closure in a timely manner. In acute wound healing, ECs in the wound’s periphery respond significantly to growth factors produced by macrophages and form vessel branches via angiogenesis towards the wound’s centre [33]. In fact, macrophages lead the ECs’ sprouting tips during early angiogenesis [66]. The molecular crosstalk between macrophages and ECs during angiogenesis is well established and macrophages produce various growth factors (e.g., VEGF, PDGF, fibroblast growth factor (FGF), TNF) that can induce ECs’ proliferation, differentiation and survival (Figure 1B (iii)) [67,68]. Simultaneously, MMPs produced by macrophages help VEGF bioavailability to create a gradient and guide endothelial sprouting (Figure 1B (iii)) [69]. It was recently found through intravital imaging that macrophages also occupy the gaps between vessel tips in cutaneous wounds [66]. Macrophages wrapped around the newly formed vessels assist in the vessels’ anastomosis and stabilisation in the later stage of angiogenesis [66]. The importance of macrophages in maintaining endothelial survival and vessel stability is functionally validated in macrophage depletion studies in which the systemic depletion of macrophages in the proliferation phase results in immature granulation, severe haemorrhages and increased endothelial apoptosis in the granulation tissue (Table 1) [2]. 

Apart from the well-established angiogenesis process, a previous study has shown the involvement of endovascular progenitors (EVPs) in wound re-vascularisation through vasculogenesis [70]. The EVPs are present in the granulation tissue one day post-injury [70]. As the wound healing progresses, EVPs differentiate into mature ECs progressively between Day 3 and Day 7, forming new blood vessels [70]. However, the regulatory role of macrophages on EVP function and behaviour during wound re-vascularisation remains unexplored.

#### 3.3.2. Macrophage Regulation of Fibroblasts and Myofibroblasts

Complete wound closure in the murine model of cutaneous repair requires myofibroblast contraction and keratinocyte re-epithelisation. During the proliferation phase, fibroblasts are recruited from the surrounding tissue. Dermal fibroblasts are activated by growth factors, such as PDGF-α, which induce fibroblast proliferation and differentiation into myofibroblasts through TGF-β signalling (Figure 1B (iv)) [33,71]. Myofibroblasts are the key drivers of wound contraction as well as collagen synthesis and deposition, which replace the provisional fibrin matrix, thus initiating the wound-closure process. Aberrant myofibroblast formation has contributed to fibrosis and excessive scar formation. Although the molecular mechanism remains incompletely defined, research studies over the years have indicated chronic inflammation as the leading cause of fibrosis development [72]. Macrophages are one of the major producers of inflammatory cytokine, and their depletion during the early and middle stages of wound healing shows reduced scar formation, indicating their involvement in fibrosis (Table 1) [2,73]. 

More recently, it has been shown that dermal CD301b^+^ macrophages enriched with PDGF-C and insulin growth factor 1 (IGF-1) regulate the proliferation of a myofibroblast subset derived from adipocyte precursors (AP) (Figure 1B (v)) [20]. The abundance of CD301b^+^ macrophages associated with reduced adipocyte-derived myofibroblast numbers in aged mice may explain the delay in wound healing during old age (Table 1) [20,58]. CD301b^+^ macrophage depletion caused defects in fibroblast/myofibroblast recruitment, reduced re-epithelisation and re-vascularisation after injury [58]. PDGF-C overexpression is a key driver of fibrosis progression in various organs [74,75,76]. Although macrophage-secreted PDGF was shown to promote wound fibrosis by inducing osteopontin expression in fibroblasts, no assessment of collagen deposition was mentioned in the study by Shook et al. Hence, the involvement of CD301b^+^ macrophages on the outcomes of scar formation and fibrosis remains elusive. 

During the past decades, the phenotypic characterisation of macrophage subpopulations and their association with diseases such as fibrosis remain a major hurdle. However, with the emerging evidence indicating the pro-fibrotic role of macrophages, it is important to discern macrophage subsets that drive fibrosis progression for the development of novel targeted therapies. 

Recently, single-cell RNA sequencing of F4/80^+^CD64^+^ macrophages in the regenerative and fibrotic microenvironment has shown that CD301b^+^ macrophages adopt a pro-regenerative characteristic while CD301b^neg^CD9^+^ macrophages are pro-fibrotic [77]. CD301b^+^ macrophages are enriched with genes involved in glycolysis, adaptive immune activation, phagocytosis and anti-inflammation [77]. In contract, CD301b^neg^CD9^+^ macrophages have an overrepresentation of pro-inflammatory and auto-immunity gene signatures [77]. Notably, a subset of CD301b^neg^CD9^+^MHCII^neg^ macrophages were shown to express the interleukin 17 receptor A (IL-17ra) and produce IL-36γ in response to IL-17 stimulation [77]. IL-36γ belongs to the IL-1 superfamily and is a driver of inflammatory responses in macrophages and fibroblasts [78]. IL-17a and IL-17ra knockout have been shown to reduce CD9^+^ macrophage numbers and prevent fibrosis, suggesting a regulatory role of IL-17a on macrophage function and fibrosis progression [77]. In fact, IL-17 is associated with various fibrotic diseases and is known to be present in hypertrophic scars and fibrotic skin from patients [79,80]. Overall, these results clearly suggest that macrophages play a significant role in wounds’ myofibroblast activity. Thus, a better understanding of myofibroblast heterogeneity as well as the macrophage populations regulating those myofibroblasts is critical for the development of macrophage-targeted therapies.

The endothelial-to-mesenchymal transition (EndMT) is an event in which endothelial cells lose specific features, such as tight junctions, and acquire the characteristics of mesenchymal cells, such as increased cell motility, alpha-smooth muscle actin expression, ECM protein expression and contractile capability. Accumulating evidence suggests EndMT as another mechanism for myofibroblast generation. Indeed, the conditional abrogation of Notch signalling in ECs results in accelerated EndMT [81], indicating the importance of Notch signalling in maintaining an endothelial cell fate. Additionally, using an endothelial-specific knockout murine model, Zhao et al. revealed the antagonistic role of Notch signalling and the transcription factor, Sox9, in cutaneous wounds [82]. Specifically, Sox9 knockout in the vasculature reduced EndMT in skin wounds by upregulating canonical Notch signalling in ECs [82]. Importantly, the EndMT process might involve transcriptional regulation at the EVP level, as Sox9 was found to be upregulated in the EVPs [70]. It is worth mentioning that Sox9 is required for ECs to complete the mesenchymal transition during the endocardial cushion formation in the development of the heart [83,84]. Coincidentally, a recent study has also shown that macrophage-specific MMP14 gene deletion results in less EndMT in a murine model of myocardial infarction via modulating TGF-β signalling [85]. It is well established that macrophage depletion in wound healing can reduce myofibroblast numbers and result in less scar formation (Table 1) [2]. Together, the above findings may suggest a role for macrophages in regulating wound-related EndMT.

#### 3.3.3. Macrophage Regulation of Keratinocyte Re-Epithelisation

Homeostasis and the regeneration of the skin epithelium are governed by several populations of epidermal progenitors residing within the basal layer and in the HF [86]. During wound re-epithelisation, the epidermal leading edge is composed of non-proliferative migrating keratinocytes followed by the proliferative epidermal progenitors [23,87,88]. Lineage-tracing studies using various epidermal stem cell murine models have shown that Krt14^+^ interfollicular epidermal (IFE) progenitors and HF progenitors, such as the Krt15^+^Lgr5^+^ bulge/hair germ, Lgr6^+^ isthmus and Lrig1^+^ infundibulum, contribute individually to wound re-epithelisation with distinct dynamics [23,87,88,89]. Although macrophages negatively regulate HF progenitor quiescence during homeostasis, they are essential for keratinocyte proliferation during wound healing. Recent findings by Ceradini et al. have uncovered the molecular crosstalk between keratinocytes and macrophages in cutaneous wound healing in which Nrf2 in Krt14^+^ basal keratinocytes transcriptionally initiates CCL2 expression to promote macrophage trafficking after wounding (Figure 1A (vi)) [90]. Infiltrating macrophages are necessary for epidermal growth factor (EGF) production in order to activate keratinocyte proliferation and hence promote wound closure (Figure 1B (vi)) [90]. Importantly, the ablation of Nrf2 from the epidermis reduced macrophage infiltration in the wound. In contrast, the topical application of CCL2 rescued the delayed wound repair in this context by elevating the number of macrophages and EGF production, hence restoring keratinocyte proliferation (Table 2) [90].

Furthermore, exosomes represent another mode of cell–cell communication with an impact on wound healing [104,105,106]. Keratinocyte-derived miRNA containing exosomes in the wound helped to promote the macrophages’ anti-inflammatory phenotype switch [104]. The in vitro treatment of the keratinocyte-derived exosome to pro-inflammatory macrophages reduced pro-inflammatory cytokine production while increasing anti-inflammatory cytokine synthesis [104]. In contrast, the delivery of siRNA targeting keratinocyte exosome miRNA release using lipid nanoparticles caused prolonged pro-inflammatory persistence in the wound [104]. Although no significant impact was observed on the wound closure rate after the exosome inhibition, the epithelial integrity was severely compromised with a reduced number of epithelial junctional proteins [104]. Based on the above studies, it is apparent that molecular signalling from the keratinocytes has indispensable effects on macrophages’ functional behaviour. Notably, the bi-directional molecular crosstalk between macrophages and keratinocytes orchestrates the re-epithelisation process. However, knowledge of the molecular pathways involved in this process is still currently limited.

### 3.4. Macrophage Regulation in Remodelling Phase (Day 7–Months)

After complete wound closure, remodelling is the final stage of wound healing that happens within weeks or months. In this process, the ECM and collagen fibres in the wounded area are re-organised into a sturdy matrix to restore the skin’s integrity [33]. Additionally, the immune infiltrates of the innate and adaptive system return to the baseline level while the myofibroblasts and the newly formed blood vessels regress via apoptosis and possibly via EndMT (Figure 1C (i) and (ii)) [81]. Macrophages contribute to vessel regression and reorganisation of the vascular bed by phagocytosing apoptotic ECs and myofibroblasts (Figure 1C (i)) [66]. Macrophages also degrade and restructure the ECM component in the wounded area [2,33]. However, their depletion at this late stage of wound healing had minimal effects on wound outcomes (Table 1) [2].

## 4. Macrophage Dysfunction in Chronic Wound

It is apparent that macrophages orchestrate the biological function of different cell types driving wound healing. Undoubtedly, macrophage dysfunction is one of the leading causes of chronic wound development. The most common types of chronic wounds are venous leg ulcers followed by diabetic foot ulcers. The underlying pathophysiological reasons for non-healing wounds are different between these. However, both ulcerations present with chronic inflammation characterised by delayed immune infiltration, myeloid cell persistence, elevated pro-inflammatory cytokines, such as IL-1β, IL-6, and TNF, increased proteolytic enzymes (e.g., MMP2 and MMP9), reduced ECM deposition, stem cell senescence (e.g., dermal fibroblasts), and increased cell death. Notably, the failure of the phenotype switching of pro-inflammatory to anti-inflammatory macrophages which delays their entry in the proliferation phase is the aetiology of the non-healing wound [3]. 

Venous leg ulcers often develop due to venous insufficiency of the lower limbs causing restricted blood movement (venous stasis) and venous hypertension [3]. Leukocytes, erythrocytes, and fibrin accumulate in the veins due to venous stasis extravasate and deprive the surrounding tissue of oxygen and nutrients, triggering the macrophages’ infiltration [3]. The engulfment of erythrocytes induced iron accumulation by macrophages resulting in unrestrained pro-inflammatory signalling, characterised by enhanced TNF, IL-12 and ROS production [3]. TNF can activate NF-κB in macrophages via a positive feedback mechanism, creating a vicious pro-inflammatory cycle in venous leg ulcers, hence contributing to chronic inflammation [107]. Additionally, ROS produced as a by-product of the iron redox reaction can cause the premature senescence of the resident fibroblasts, degrade tissue and hence impair wound healing in venous leg ulcers [3].

Although diabetic foot ulcers are also associated with hyperinflammation, the mechanism of initiation is different. High blood glucose levels in diabetic patients induce the production of advanced glycation end products (AGEs), glycated lipid and protein products due to the exposure to sugar [108]. AGEs bind to the receptor for advanced glycation end products (RAGEs) on macrophages and induce pro-inflammatory factors via NF-κB signalling [108,109,110]. The hyperglycaemic microenvironment further induces low-grade systemic inflammation and altered myeloid functions due to changes in the haematopoietic stem cell niche during homeostasis [111]. Prominently, bone marrow progenitors from *db/db* mice are biased towards myeloid commitment during homeostasis [112]. Myeloid progenitors from diabetic mice are intrinsically sensitized to pro-inflammatory macrophage polarisation and display reduced phagocytic activity, leading to neutrophil accumulation in diabetic wounds [53,54,113]. In conjunction, circulating myeloid cells from diabetic patients also have a reduced phagocytic capability in vitro [114], reducing efferocytosis and the transition to an anti-inflammatory phenotype [51]. The negative effect of AGEs on wound healing was further confirmed by an anti-RAGE antibody treatment in which the blocking of the AGEs/RAGE signalling improved diabetic wound healing [115]. Overall, exaggerated proinflammatory macrophage activity is one of the leading causes of chronic wounds, characterised by the overactivity of the NF-κB and its downstream signalling. Therefore, targeting pro-inflammatory macrophages, their cytokines or NF-κB activity, may be a plausible therapeutic strategy for chronic wounds. To date, it remains elusive if the macrophage dysfunction observed in chronic wounds of various aetiologies is related to any specific subset of macrophages. This information would be paramount for developing targeted therapies for each chronic wound type.

## 5. Strategies to Modulate Wound Inflammation and Macrophage Activity

Given the diversity and complexity of macrophage populations in skin wounds, it seems unlikely that a single therapeutic strategy can be universally applied to improve healing at every stage. Many studies have therefore attempted to dissect the molecular mechanisms involved in inflammatory macrophages that are dysregulated in chronic wounds.

### 5.1. NF-κB Signalling–A Friend or Foe of Wound Healing

NF-κB is the central inflammatory mediator in early wound healing. It is required for the induction of many inflammatory genes (e.g., TNF, IL-1β, IL-6, IL-12p40, IFN-γ and Cyclooxygenase-2) and antimicrobial peptide in different cell types, including macrophages [107]. Multiple pro-inflammatory factors, mainly toll-like receptor (TLR) ligands, IL-1β and TNF can synergistically induce NF-κB signalling with some of the downstream targets forming positive or negative feedback mechanisms, regulating NF-κB activity and other inflammatory pathways [100]. Importantly, the activation/downregulation of the NF-κB signalling needs to be controlled precisely in a spatiotemporal manner as it plays a central role in orchestrating the innate/adaptive immune responses during skin homeostasis and regeneration. The dysfunction of NF-κB either due to hyper-reactivity or insensitivity is associated with inflammatory skin diseases (e.g., psoriasis and atopic dermatitis) [116] and can prominently affect wound-healing outcomes. 

#### 5.1.1. Toll-like Receptors (TLRs) and MyD88 Adaptor Protein

The binding of LPS (a component of bacterial membranes) or free RNAs to TLR4 or TLR3 induces canonical NF-κB signalling via the *MyD88* and TIR-domain-containing adapter-inducing interferon-β (TRIF) adaptor proteins, respectively [93,94]. Attenuating the TLRs mediated NF-κB signalling by knocking out its downstream *MyD88* adaptor protein globally or altering TRIF by siRNA significantly delaying wound healing with altered macrophage infiltration and polarisation, indicating NF-κB as a key modulator of macrophage functions (Table 2) [91,92,93]. *MyD88*^−/−^ wounds also had reduced vessel growth due to diminished TNF, hypoxia-inducible factor (HIF)-1 and VEGF production, probably from altered macrophage activity (Table 2) [91]. Similarly, the conventional knockout of the upstream inducer, TLR3 and TLR4, severely impaired wound closure (Table 2) [93,94]. The reduced production of the NF-κB downstream targets, such as IL-6, TNF, CXCL2, CCL3 and CCL2, in these knockout mice led to altered neutrophil and macrophage infiltration, as well as reduced keratinocyte proliferation and migration via direct and indirect mechanisms [93,94]. In some aspects, these observations recapitulate macrophage depletion studies.

#### 5.1.2. IL-1β/IL-1R1 Signalling

The IL-1β/IL-1R1 signalling axis is another major upstream regulator of the NF-κB transcription factor also via the *MyD88* and TRAF6 adaptor proteins. Importantly, IL-1β can positively self-regulate after NF-κB activation through the NLRP3 inflammasome complex which consists of NLRP3, ACS and Caspase 1 [117]. NF-κB induces NLRP3 inflammasome priming and activation, and in turn, the NLRP3 inflammasome promotes IL-1β maturation by cleaving the IL-1β precursor, pro-IL-1β [117]. IL-1β activity further promotes NF-κB signalling, creating a vicious cycle of pro-inflammatory signalling [117]. During early wound healing, macrophages and keratinocytes are the main producers of IL-1β required to sustain the inflammatory responses [92,100]. Mice with global IL-1β knockout displayed delayed wound closure, suggesting the importance of IL-1β signalling in the healing process (Table 2) [92]. 

Although NF-κB signalling is required for effective wound closure, its exacerbated activity correlates with delayed wound repair. Mice with the gene knockout of the IL-1 receptor antagonist (IL-1ra) showed attenuated wound healing characterised by exaggerated neutrophil infiltration, reduced collagen deposition and neovascularization [118]. Increased NF-κB signalling evidenced by enhanced and prolonged p65 nuclear translocation in the *Il-1ra*^−/−^ wound was related to increased IL-1β, TNF, macrophage inflammatory protein 2 (MIP-2), and decreased TGF-β and VEGF signalling [118]. The detrimental effects of exacerbated IL-1β signalling in delayed wound healing were further supported by clinical and preclinical observations of diabetic chronic wounds. Chronic wounds from diabetic patients and *db/db* mice showed the overproduction of IL-1β in macrophages [119,120]. Anti-IL1β-antibody-treated and *Il-1r1*^−/−^ diabetic skin wounds showed accelerated wound closure, and resulted in improved collagen deposition, IGF-1 and TGF-β production, and reduced IL-1β and TNF expression (Table 3) [119,120]. Similarly, the topical delivery of IL-1ra enhanced neutrophil clearance and macrophage recruitment in diabetic wounds, promoting a pro-healing microenvironment for wound repair (Table 3) [119]. 

Based on the above preclinical data, it is apparent that modulating NF-κB or its downstream targets in the wound can fine-tune macrophage kinetics and polarisation to help improve chronic wound healing. With no surprise, a recent phase III clinical study using a novel macrophage-regulating drug, ON101, on patients’ diabetic foot ulcers showed improved wound healing [121]. More than 60% of the patients completed wound healing within the 16 weeks of trial compared to only 35% in the placebo group [121]. The macrophage-regulating drug consists of an anti-inflammatory compound, PA-F4, which can suppress the macrophages’ NF-κB/NLRP3/IL-1β inflammasome pathway, and a natural pro-healing compound derived from the plant *Centella asiatica*, which can help with collagen synthesis, fibroblast proliferation and keratinocyte migration [122,123]. This formulation exerts a synergic effect in balancing the pro-inflammatory to anti-inflammatory macrophage ratio in chronic wounds, allowing them to progress into the proliferation phase of wound healing [121,122].

#### 5.1.3. IL-1β and IL-17A Signalling Crosstalk 

Various studies have also shown the synergistic role of IL-1β and IL-23 in inducing IL-17A production during skin inflammation and repair [100,124,125]. IL-17A is required for neutrophil recruitment and keratinocyte proliferation and is associated with hyperproliferative skin diseases, such as psoriasis and skin cancer (Table 2) [77,96,126]. In murine skin, IL-17A is largely produced by various subsets of T cells, including γδT cells, such as the dendritic epidermal T cells (DETC), dermal Vγ4 T cells, and T helper 17 (Th17) cells [97,99,100,103]. Utilising the *Il17a*^−/−^ mice and *Tcrd*^−/−^ mice that lack the DETC, MacLeod et al. has showed that DETC-derived IL-17A is required for keratinocytes’ antimicrobial peptide production and effective wound closure, as IL-17A deficiency resulted in wound-healing defects and reduced keratinocyte proliferation (Table 2) [97]. Correspondingly, another study by Chen et al. has shown that IL-17A induced Lrig1^+^ HF progenitors’ proliferation and migration into the IFE during homeostasis and wound healing [102]. Abolishing IL-17 signalling in the Lrig1^+^ HF progenitors, which contributes to wound re-epithelisation, showed impaired wound closure (Table 2) [102]. Mechanistically, Chen et al. demonstrated that IL-17A activates IL-17RA coupled to epidermal growth factor receptor (EGFR) in keratinocytes, driving Lrig1^+^ progenitor cells’ expansion and migration to the IFE via p-ERK5 [102]. A recent study by Konieczny and colleagues also delineated IL-17A as the upstream activator of the mTOR/HIF1α pathways, regulating Krt14^+^ keratinocyte glycolysis activity during re-epithelisation [103]. Keratinocyte HIF1α expression could only be specifically activated by RORγt^+^ γδT cell-derived IL-17A and helped fuel energy for the migrating epithelium at the wound front [103]. Depleting RORγt^+^ γδT cells or the inactivation of the IL-17RC, HIF1α or mTOR in keratinocytes reduced epithelial tongue migration, hence delaying wound closure (Table 2) [103]. Overall, the independent research findings above highlight the importance of IL-17A signalling and its diverse implication on keratinocyte subpopulations during wound re-epithelisation. 

Nonetheless, Rodero et al. and Lee et al. have shown that chronic wounds from obese mice (*ob/ob*) or wounds from animals maintained outside pathogen-free conditions have elevated level of γδT cells and IL-17A cytokine in an IL-23 dependent way [98,99]. Neutralising IL-17A or IL-23 using antibodies to IL17-A or P19 or *Il-17a*^−/−^ knockout in obese mice improved chronic wound healing by promoting macrophage anti-inflammatory phenotype switching and reduced neutrophils infiltration (Table 2 and Table 3) [98,99]. Interestingly, Li et al. has recently demonstrated that different cell sources and concentrations of IL-17A played diverse immunoregulatory roles in wound healing, explaining the contradictory observations reported in the previous literature. A moderate dose of anti-IL17A antibody treatment was shown to improve wound closure whereas a high dose of recombinant IL-17A (rIL-17A) treatment delayed wound repair (Table 2) [100]. Furthermore, an in-depth mechanistic study revealed the inhibitory role of Vγ4 T cell-derived IL-17A on DETC’s IGF-1 production via the IL-1β/*MyD88*-dependent pathway (Figure 1B (vii)) [100]. The inhibitory mechanism involves the IL-17A/IL-1β/IGF-1 signalling cross-regulation between the Vγ4 T cells, keratinocytes and DETC (Figure 1B (vii)). Of note, DETC is the exclusive source of IGF-1 in the epidermis [127]. Importantly, IGF-1 helps to maintain epidermal cell survival during skin homeostasis and promotes keratinocyte proliferation and migration during wound healing [127]. Together, these findings support the notion that a low level of IL-17A is necessary for effective wound healing, particularly in preventing pathogen colonisation, as well as promoting keratinocytes’ proliferation, migration, and innate immune cells’ recruitment. However, uncontrolled IL-17A activity has a negative impact on the wound-healing process by intensifying the inflammatory response via pro-inflammatory cytokines’ production as evident in chronic wounds. Most importantly, case reports of the use of Ustekinumab (anti-IL23 monoclonal antibody) and Secukinumab or Ixekizumab (anti-IL17A monoclonal antibody) treatments in pyoderma gangrenosum (a skin ulceration condition as a comorbidity of other chronic inflammatory diseases) show improved wound healing without compromising keratinocyte function [128,129,130,131].

Apart from the γδT cells, Th17 cells also contribute to IL-17A production. Although the regulatory function of Th17 cells in cutaneous wound healing remains obscure, exacerbated Th17-mediated responses have been regarded as the molecular driver in other chronic skin inflammatory diseases (e.g., psoriasis) [132]. Both chronic non-healing wounds and psoriasis manifest with increased Th17 cells and pro-inflammatory macrophages, and reduced regulatory T cells (Treg) cells [99,132]. Keeping Th17 and Treg cell numbers balanced is crucial to maintaining skin homeostasis. Specifically, Treg cells help prevent excessive immune activation or autoimmune responses by producing anti-inflammatory cytokine (e.g., IL-10 and TGF-β) and modulating the immune activity of other innate cells (e.g., macrophages) and CD4^+^ and CD8^+^ lymphocytes via direct or indirect mechanisms [133]. Activated Treg cells in wounds express high levels of cytotoxic T-lymphocyte-associated antigen 4 (CTLA-4) and inducible T cell costimulatory (ICOS) [134]. ICOS/ICOSL signalling regulates Foxp3^+^ Treg development, survival and immunosuppressive function. ICOS^−/−^ mice have reduced Foxp3^+^ Treg cell numbers [135,136]. Abrogating ICOS signalling or Treg depletion during cutaneous wound healing negatively affects wound healing [134,137]. Treg played a significant role in suppressing CD4^+^ and CD8^+^ T cells’ activation, proliferation and cytokine production. Treg depletion during wound healing perturbed the fine balance and increased the infiltration of CD4^+^ and CD8^+^ T cells that produced IFN-γ [134]. The number of pro-inflammatory macrophages was also increased, likely due to the pro-inflammatory cytokine milieu [134]. Notwithstanding, CD8^+^ T cells were shown to promote macrophage recruitment and pro-inflammatory polarization in obese and high-fat-diet mice [138]. It is plausible that both Treg and CD8^+^ T cells play concomitant roles in modulating macrophage kinetics in wound healing.

In contrast to the Treg depletion model, ICOS knockout mice have reduced wound CD3^+^ T cells, macrophages and pro-inflammatory cytokines’ production (e.g., IL-1, IL-6, TNF, etc.), suggesting an essential role of ICOS in wound healing, notably the immunoregulatory crosstalk between the innate and adaptive system [137]. The ligand for ICOS, ICOSL, is expressed mainly on antigen-presenting cells (e.g., macrophages, dendritic cells), B cells, and some other non-immune cells (e.g., activated endothelial cells, fibroblasts and keratinocytes). Topical treatments of the soluble form of ICOS recombinant protein (ICOS-Fc) on ICOS^−/−^ wounds improved wound repair by enhancing keratinocyte, fibroblast, myofibroblast and endothelial function [139]. The activation of the ICOS/ICOSL signalling also increased M2 macrophage migration, restored IL-6 and reduced TNF expression [139]. Together, these results indicated that the co-stimulatory ICOS/ICOSL signalling has pleiotropic effects on cutaneous wound healing. However, the underlying mechanism involved still requires further investigation, particularly how the Treg and macrophages cross-regulate each other in the context of wound healing.

Programmed cell death protein 1/programmed death-ligand 1 (PD-1/PD-1L) signalling is well established for its immunosuppressive role on T cells in autoimmune diseases and cancer settings. In recent years, many studies have looked beyond the intrinsic roles of PD-1/PD-1L on leukocytes and focused on its crosstalk with other immune and non-immune cells. Lately, Piao et al. have conducted an intricate study demonstrating the molecular mechanism behind Treg and effector T cells’ (Teff) migration mediated by the PD-1/PD-1L signalling axis [140]. PD-1L expressed on lymphatic endothelial cells is required for Treg and Teff to migrate across the endothelium [140]. Notably, Treg expressed a higher level of PD-1 and the engagement of PD-1/PD-1L only allows the transmigration of Treg selectively [140]. On the other hand, CD80/PD-1L specifically controls Teff cell migration [140]. To note, PD-1L activation in the lymphatic endothelial cells changes the expression of VCAM, and it was mediated downstream of the NF-κB and ERK signalling [140]. Overall, the above study indicates that PD-1L helps to regulate Treg and Teff cells’ recruitment and targeting PD-1/PD-1L can influence the immune landscape in various diseases, such as chronic wound healing. To this end, it was shown that diabetic non-healing wounds had downregulated PD-1L [141]. Topical applications of exogeneous PD-1L enhanced the closure of diabetic wounds in parallel with the resolution of hyperinflammation and reduced macrophage cell numbers [141]. Interestingly, the activation of PD-1L also promoted keratinocyte proliferation and migration, highlighting the diverse role of PD-1/PD-1L in non-immune cells [141]. PD-1 is known to be expressed in myeloid cells, and myeloid-specific PD-1 knockout increased CD8^+^ T cells that expressed IFN-γ and IL-17 in a melanoma cancer model [142]. Nevertheless, the kinetics of various T cell subsets and macrophage polarization states after PD-1L treatment remain unexplored in a cutaneous wound healing setting.

#### 5.1.4. TNF/TNFR1 Signalling

TNF/TNFR signalling regulates many facets of cell function, including cell survival, proliferation, differentiation and apoptosis. The engagement of TNF to its receptors TNFR1 and TNFR2 triggers a cascade of events leading to the activation of various signalling complexes involved in the NF-κB, mitogen-activated protein kinase (MAPK), and cell death signalling pathways [143]. Under most circumstances, TNF induces NF-κB pro-inflammatory signalling. However, NF-κB dysregulation can divert TNF-mediated signalling to activate RIPK1-mediated cell death, namely apoptosis (RIPK1/FAAD/Caspase-8-dependent) and necroptosis (RIPK1/MLKL/RIPK3-dependent) [144,145,146]. Apoptosis is a programmed cell death in which the cell apoptotic bodies are cleared by professional phagocytes, such as the macrophages [147]. Apoptosis was once thought to be non-immunogenic, unlike necroptosis which induces cell lysis, releasing its cytoplasmic content (e.g., DAMPs), and as a result triggering pro-inflammatory responses [148]. However, it is now clear that the apoptosis and necroptosis pathways are highly complex, and both could trigger inflammation. However, the role of these pathways and biological roles in human inflammatory diseases is not fully understood. 

TNF/TNFR signalling played vital roles in skin homeostasis and various research studies identified TNF as a molecular driver of chronic skin inflammatory diseases (e.g., psoriasis and hidradenitis suppurativa) [149,150,151]. Mice lacking IKK2, a kinase required for NF-κB activity, paradoxically resulted in the development of a TNFR1-dependent psoriasis-like phenotype [152,153]. The skin lesion development in these mice was associated with macrophage accumulation in the skin, and the liposome-mediated delivery of clodronate ameliorated the skin lesions [154]. Recently, Kumari et. al has shown that the NF-κB inhibition in epithelial cells induces TNF-mediated, RIPK1 kinase-dependent apoptosis and necroptosis, which trigger skin inflammation in these mice [145,146,152]. These results indicate that epidermal TNFR1/NF-κB signalling is pivotal in maintaining keratinocyte survival. Keratinocyte cell death activates inflammatory responses, which promote innate immune cell infiltration, particularly the macrophages and neutrophiles, further highlighting the role of immunoregulatory crosstalk between the keratinocytes and the myeloid cells in maintaining skin homeostasis. Interestingly, Nrf2 and NF-κB were shown to negatively regulate each other by counterbalancing the oxidative stress and inflammatory response [155]. The keratinocyte-specific ablation of Nrf2 during wound healing significantly delays wound closure and alters macrophages’ recruitments via reduced Ccl2 expression (Table 2) [90].

TNF is produced from macrophages and neutrophils after injury, and its expression is highest during the early phase of wound healing and returns to baseline after the proliferation phase [101]. However, patients with venous leg ulcers and diabetic foot ulcers have increased NF-κB activity and TNF expression in the serum and wound exudate. The NF-κB activity and TNF level in the wound negatively correlated with the healing status, suggesting a detrimental role of TNF in chronic wound pathogenesis [156]. The same observation was reported in the cutaneous wound of murine type I and type II diabetic models [157]. It is noteworthy that chronic diabetic wounds were commonly presented with increased fibroblasts, keratinocytes and endothelial cell death [158,159,160]. However, the cell type-specific role of TNF-mediated functions in wound-healing responses are poorly understood. In vitro, TNF induces fibroblasts and endothelial cell death [161,162,163]. Attenuating TNF signalling reduces neutrophil and macrophage numbers and fibroblasts’ apoptosis, as well as restores fibroblast proliferation and collagen synthesis, eventually leading to improved wound closure (Table 2) [101,160]. Most importantly, clinical trials using the anti-TNF neutralising antibodies, Adalimumab and Infliximab, on venous leg ulcers that were previously unresponsive to existing treatment have improved patients’ healing outcomes [164,165]. Although preclinical and clinical evidence has shown the therapeutic potential of anti-TNF antibody on chronic non-healing wounds, the immunomodulatory mechanistic roles of TNF signalling as well as apoptosis and necroptosis in chronic wound progression remain elusive. 

Overall, based on the research findings presented above, it is evident that keratinocyte-specific NF-κB signalling is a crucial regulator of skin immune homeostasis. It is plausible that NF-κB signalling dysregulation in patients predisposed to chronic wound development, such as diabetes, might have responded differently to TNF and investigating the TNF-induced cell death associations and functions in chronic wounds is crucial to understanding the underlying mechanisms. As such, it is important to delineate the role of the key modulators involved in cell death, apoptosis and necroptosis, using murine models in combination with human in vitro and ex vivo studies. This knowledge can help establish the causative link between skin homeostasis and the pathogenesis of chronic non-healing wounds. The identification of key molecular targets that cause cell death in non-healing wounds can aid in new prophylactic and therapeutic developments. 

#### 5.1.5. Microbiome 

As mentioned previously, during the inflammatory phase of wound healing, macrophages play a pivotal role in pathogen clearance to prevent wound infection. While chronic wounds have excess neutrophil and macrophage recruitment, it frequently presents with persistent infection and biofilm accumulation [166]. One of the prominent bacteria in human chronic wounds of various etiologies has been identified by 16S rDNA pyrosequencing as *Staphylococcus aureus* (*S. aureus*) [167]. *S. aureus* in biofilm is less efficient in eliciting chemokine (CXCL-1, CXCL-8 and granulocyte-macrophage colony-stimulating factor (GM-CSF)) and proinflammatory cytokines production (eg, IL-1 and IL-6) in human keratinocytes compared to *S. aureus* in suspension [168]. While emerging evidence has supported the importance of keratinocytes in macrophage recruitment during cutaneous wound repair, the formation of *S. aureus* biofilm can further compromise macrophage function in pathogen clearance. Importantly, *S. aureus* can also hide within keratinocytes as a mechanism to escape immune surveillance and promote bacterial dissemination [169]. Antibiotic resistance further complicates care for chronic wounds colonized by *S. aureus* [170]. Thirty-three percent of the chronic wounds in Wolcott et al.’s study harboured methicillin-resistant *Staphylococcus* (resistant to β-lactam antibiotics) [167] and to date, *S. aureus* drug resistance remains a major healthcare challenge worldwide [171]. 

In recent years, several pieces of evidence have indicated the prominent effect of commensal microbiota in maintaining skin homeostasis and tissue regeneration [172]. Other species of *Staphylococcus*, such as the *S. hominis*, *S. capitis* and the *Cutibacterium acnes*, were found to produce potent antibiotics or antagonists that can inhibit *S. aureus* growth [173,174,175]. Peptides produced by these commensal bacteria species can also induce host immune responses, particularly the keratinocytes, to produce antimicrobial peptides and chemokines via the TLR/*MyD88* pathway [176]. Recently, emerging evidence has shown the pivotal role of microbiota in modulating the host’s inflammatory process [92,176]. A study conducted by Wang et al. comparing murine cutaneous wound healing in different grades of animal facilities (e.g., germ-free vs. specific pathogen-free vs. normal environment) has shown that commensal bacterial diversity on the skin can significantly affect the immune cells’ activation and quality of skin regeneration (Table 2) [92]. Specifically, in a murine acute cutaneous wound healing model, Wang et al. has demonstrated that *S. aureus* can induce IL-1β/*MyD88* signalling in keratinocytes, and affect the quality of wound regeneration [92]. Notably, Neosporin, an antibiotic treatment on patients’ wounds, reduced *Staphylococcus* species abundance and delayed wound re-epithelisation by diminishing *IL-1β*, *TNF*, *WNT7B*, *EGFR*, *PCNA*, and *BMP6* production [92]. Nonetheless, the unresolved IL-1β production, such as that observed in chronic wounds, induced a vicious inflammatory cycle via the NF-κB signalling. Overall, the above findings have highlighted the immunoregulatory role of the microbiota in safeguarding the skin’s immunity and tolerance against opportunistic pathogens. The overuse of antibiotics can increase resistance, disturb the microbiota niches and reduce commensal bacteria α-diversity. As a consequence, many research studies have linked unbalanced skin microbiota compositions as one of the causes of skin inflammatory and autoimmune diseases, such as atopic dermatitis [173,175]. 

With emerging evidence elucidating the commensal microbiota’ modulation on the host’s immune system, the re-establishment of the microbiota niches appears to be a viable option to treat antibiotic resistance. Interestingly, modulating microbiota from wounds can help chronic wound closure. Secretomes from an anti-inflammatory commensal gut microbiome can downregulate NF-κB induction, dampen the pro-inflammatory response, and improve wound closure in *ob/ob* mice [177]. This study has further highlighted microbiome regulations on the host’s immune response and has shown the feasibility of using microbial compounds to change the chronic wound microenvironment [177]. 

## 6. Conclusions

In the past decade, there has been extraordinary progress in our understanding of macrophages’ functions in the skin wound-healing process and their relevance to non-healing wounds. This review provides insights into the immunoregulatory roles of macrophages in acute and chronic non-healing cutaneous wounds in mice and humans by emphasizing their interaction and molecular crosstalk with other major cell types. Particularly, macrophages are able to modulate the activity of the other cells, including neutrophils, fibroblasts, and endothelial cells, among others. Cytokines that regulate their activity (e.g., IL-1β, IL-17, TNF, etc.) could represent an untapped range of molecular targets to improve wound inflammation especially in the context of chronic wounds. Downstream of many of these cytokines, NF-κB signalling is a major regulator of macrophage infiltration and phenotypic polarization. Unresolved NF-κB signalling is a common cause of hyperinflammation and pro-inflammatory macrophages’ persistence in chronic non-healing wounds. The last part of this review highlights the various upstream regulators of the NF-κB signalling and their importance in wound healing, particularly their pleiotropic effect on keratinocyte proliferation and myeloid cell chemotaxis. It is apparent that the phenotypic consequence of NF-κB dysregulation in wound healing extends beyond cytokine/chemokine gene expression alterations and also affects macrophages’ functions and their regulatory signalling crosstalk with various cell types. Considering the different roles macrophages play in the various wound healing stages and their regulation of other cell types, modulating NF-κB or its downstream signalling might purport as an alternative therapeutic approach to fine-tune macrophage functions at a variety of stages during wound healing.

## Figures and Tables

**Figure 1 biomolecules-12-01659-f001:**
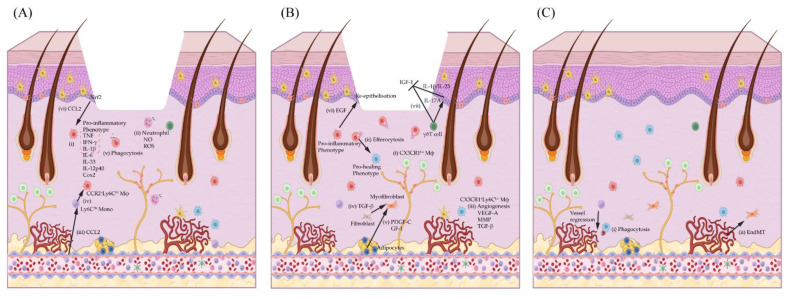
(**A**) Signalling crosstalk between macrophages (Mϕ) with other cell types during the inflammatory phase of cutaneous wound healing. (i) Nuclear factor kappa-light-chain enhancer of activated B cells (NF-κB) is major regulator of pro-inflammatory cytokines in macrophages (e.g., tumour necrosis factor alpha (TNF-α), interferon (IFN)-γ and interleukin (IL)-1β). (ii) Neutrophils produce nitric oxide (NO) and reactive oxygen species (ROS) to prevent pathogen colonisation. (iii) (C-C motif) ligand 2 (CCL2) released by tissue-resident cells recruits Ly6C^Hi^ monocytes into the wound during the inflammatory phase. (iv) Ly6C^Hi^ monocytes differentiate into (C-C motif) chemokine receptor 2 (CCR2^+^)Ly6C^Hi^ Mϕ, releasing pro-inflammatory cytokines. (v) Macrophages help to remove pathogen via phagocytosis. (vi) Nrf2 mediates keratinocytes’ CCL2 production which is required for CCR2^+^ monocyte recruitment. (**B**) Signalling crosstalk between macrophages and other cell types during the proliferation phase of cutaneous wound healing. (i) (C-X3-C motif) chemokine receptor 1 (CX3CR1)^Lo^ Mϕ guide nerve sprouting during wound repair and acquire high CX3CR1 expression progressively after injury. (ii) Efferocytosis of neutrophils induce macrophage anti-inflammatory phenotype switch. (iii) CX3CR1^+^Ly6C^Lo^ Mϕ (pro-healing phenotype) produce anti-inflammatory cytokines and growth factors, such as vascular endothelial growth factor (VEGF), matrix metalloproteinase (MMP) and transforming growth factor (TGF)-β that help to dampen inflammation and promote angiogenesis and wound contraction. (iv) TGF-β is a crucial regulator of fibroblasts/myofibroblasts. (v) Platelet-derived growth factor C (PDGF-C) and IGF-1-producing CD301b^+^ Mϕ promote the proliferation of adipocyte-derived myofibroblasts during wound healing. (vi) Infiltrating F4/80^+^ macrophages produce epithelial growth factor (EGF) to promote keratinocyte re-epithelisation. (vii) Insulin-like growth factor 1 (IGF-1) from dendritic epidermal T cells (DETC) also helps to promote re-epithelisation, while IL-17A from γδT cells and IL-1β/IL23 produced by keratinocytes negatively regulate DETC IGF-1 production. (**C**) Macrophages regulate vessel regression during the remodelling phase of cutaneous wound healing. (i) Macrophages remove apoptotic endothelial cells via phagocytosis. (ii) Emerging evidence suggests that endothelial cells might undergo endothelial-to-mesenchymal transition (EndMT) during cutaneous wound healing.

**Table 1 biomolecules-12-01659-t001:** Implications of myeloid/macrophage depletions on wound-healing outcome.

Model	Wound Type	Strain	Age (Weeks)	Functions/Phenotype	Method	Reference
*Ccr2* KO	Skin wound (4 mm)	C57BL/6	20–32	**CCR2 is required for CCR2^+^Ly6C^+^ Mϕ recruitment.** Knockout (KO) model Impaired wound healing in all stages. Reduce Ly6C^Hi^ monocytes’ infiltration into the wounds. Reduce pro-inflammatory cytokine (TNF, IL-1β and iNOS) in *CCR2* KO Mϕ. Similar percentage of neutrophiles in WT and *CCR2* KO wounds. Adoptive transfer Rescued wound-healing defect in *CCR2* KO wound.	*CCR2* global knockout and adoptive transfer of WT CD11b^+^ cells into *CCR2* KO wounded mice	[18]
*Cx3cr1* KO	Skin wound (4 mm)	C57BL/6	8–10	**CX3CR1 is required for CX3CR1^+^Ly6C^neg^ Mϕ recruitment.** Knockout model Delayed wound closure No difference in neutrophil and CD3^+^ T cell numbers Reduced F4/80^+^ macrophages Reduced collagen deposition and granulation tissue formation Deficient in myofibroblasts and TGF-β production Reduced blood vessel density and VEGF expression BM transplant study Delayed healing in wound reconstituted with *Cx3cr1* KO BM cells Reduced macrophage numbers and collagen deposition in wounds reconstituted with *Cx3cr1* KO BM cells	*Cx3cr1* global knockout and BM transplant of WT or *Cx3cr1* KO BM cells into WT or *Cx3cr1* KO wounded mice	[19]
Anti-CX3CR1 antibody	Skin wound (4 mm)	C57BL/6	8–10	**Neutralising CX3CR1 prohibit CX3CR1^+^Ly6C^neg^ Mϕ recruitment.** Impaired wound closure and collagen deposition	Neutralising antibody	[19]
*LysMCre-DTR*	Skin wound (5 mm)	C57BL/6	10–12	**Myeloid cell depletion caused impaired wound closure.** Early stage (DT injection from Day 2 to Day 4) Day 5: Delayed wound closure characterized by reduced re-epithelisation, angiogenesis, contraction, and epidermal and dermal proliferation Day 10: Neo-epithelium appeared immature and fragile; reduced granulation tissue and wound contraction Day 14: Reduced scar tissue Mid-stage (DT injection from Day 4 to Day 8) Day 7: Delayed wound closure Day 10: Immature granulation tissue showed severe haemorrhage due to EC damage and reduced wound contraction Diminished TGF-β and VEGF-A production Late stage (DT injection at Day 9 to Day 14) Day 14: No microscopic difference in wound closure rate and histological analysis	Mo/Mϕ depletion	[2]
*Mgl2—DTR/GFP* (CD301b)	Skin wound (4 mm)	C57BL/6	7–9	**CD301^+^ Mϕ regulates adipocyte-derived myofibroblasts proliferation**. Delayed re-epithelisation and re-vascularisation Reduced CD26^Low^ adipocyte precursors’ (AP) proliferation but not CD29^Hi^ and CD29^Low^ AP	CD301b^+^ Mϕ depletion	[20]
CSF1R inhibitor (BLZ945) andClodronate liposomes	Ear punch (2 mm)	C57BL/6	6–20	**Mϕ depletion impaired nerve regeneration and sprouting.** BLZ945 Reduction in nervous network Clodronate liposomes Short and undirected axons’ network	Mϕ depletion	[8]
OVA-coated nanoparticles adsorbed with DT (DT-OVA-NP)	Skin wound (5 mm)	C57BL/6	8–12	**Selective depletion of perivascular Mϕ negatively affects wound healing.** Delayed wound-healing characterized by reduced granulation tissue formation, angiogenesis, and re-epithelisation Defective collagen deposition Disordered myofibroblasts distribution	Perivascular Mϕ depletion	[6]

**Table 2 biomolecules-12-01659-t002:** Immunoregulatory role of different cytokines and pro-inflammatory signalling on wound healing.

Model	Wound Type	Strain	Age (Weeks)	Functions/Phenotype	Method	Reference
*MyD88 KO*	Skin wound (10 mm)	C57BL/6	10	**Gene knockout of adaptor protein *MyD88*, which is required for NF-κB signalling, impairs wound healing and reduces growth factor expression.** Delayed wound closure, increases F4/80^+^ Mϕ and reduces blood vessel density Reduced Hif1-α and Vegf mRNA expression	Global *MyD88* KO	[91]
*MyD88 KO*	Skin wound (1.44 cm^2^)	C57BL/6	3	**Abrogation of NF-κB signalling impairs wound healing outcome and affects host microbiota diversity.** Delays wound healing Lower anti-inflammatory macrophages and γδT cells Reduces WIHN Lower microbiota α-biodiversity	Global *MyD88* KO	[92]
*Tlr3* KO	Skin wound (4 mm)	C57BL/6	8–12	**Gene knockout of *Tlr3* affects chemokine expression and myeloid cell recruitment.** Delayed wound closure, reduced granulation formation, neovascularization, re-epithelialisation Defective recruitment of neutrophils and Mϕ No difference in CD3^+^ T cells’ infiltration Reduced CXCL2, CCL3, and CCL2 expression	Global *Tlr3* KO	[93]
*Tlr4 KO*	Skin wound (3 mm)	C3H/HeJ	6–8	**Gene knockout of *Tlr4* affects pro-inflammatory cytokines production and reduces keratinocytes’ proliferation.** Delayed wound healing in the early stage due to reduced keratinocyte proliferation Altered neutrophils, Mϕ and CD3^+^ T cell numbers Reduced IL-1β and EGF expression, and delayed IL-6 expression No difference in TNF production	Global *Tlr4* KO	[94]
*Il-1β KO*	Skin wound (1.44 cm^2^)	C57BL/6	3	**IL-1β is required for wound healing and promotes wound-induced hair neogenesis (WIHN).** Delayed wound healing Reduced WIHN	Global *IL-1β* KO	[92]
*Il-1r KO*	Skin incisional, 6 mm excisional and PVA implant	C57BL/6	8–12	**IL-1 signaling regulates pro-inflammatory cytokine expression and plays different regulatory roles in various wound types.** PVA implant Reduced fibrosis Lower levels of IL-6 at all times and reduced TGF-β and VEGF at Day 1 wound Similar level of TNF, IL-1α, IL-10, IL-12p70, IFN-β, and CCL5 Incisional wound Improved wound healing Reduced infiltration of inflammatory cells Diminished epidermal thickening Decreased area and depth of scarring Excisional wound No different in rate of wound closure Contained fewer inflammatory cell infiltrates Reduced epidermal thickness and scarring size	Global *Il-1r* KO	[95]
*Il-17a* KO	Skin wound (3 mm)	C57BL/6	-	**IL-17A is required for neutrophil recruitment and modulates collagen production in the granulation.** Knockout model Day 1: Reduced neutrophils number Day 2: Increased TGF-β production Day 3: Improved wound healing with increased COL1A1 and reduced COL3A1 expression Day 5: Improved wound healing with well-developed granulation tissue and increased myofibroblast number + rIL17 treatment on WT wound Day 1: Increased neutrophils’ recruitment Day 3: Delayed wound healing with reduced collagen deposition	Global *Il-17a* KO and administration of rIL17A on WT wound	[96]
*Il-17a* KO	Skin wound (6 mm)	C57BL/6	6–8	**IL-17A/IL-1β/IL-23 forms a positive feedback loop in the epidermis around the wound and negatively regulates IGF-1 production.** Knockout model Defective in wound repair but increases IGF-1 expression in the epidermis Reduced IL-1β and IL-23 production in the epidermis + rIL-1β and/or rIL-23 on WT wound Delayed wound repair Reduced DETC IGF-1 production Enhanced NF-κB and STAT3 nuclear translocation + anti-rIL-1β and anti-rIL-23 antibody on WT wound Improved wound closure Increased DETC IGF-1 production	Global *Il-17a* KO	[97]
*Il-17a* KO	Skin wound (6 mm)	C57BL/6	8–12	**Inhibition of IL-17A improves wound healing with increased percentage of pro-healing Mϕ** Accelerated wound closure Reduced pro-inflammatory Ly6C^+^MHCII^neg^ Mϕ and increased Ly6C^neg^MHCII^+^ anti-inflammatory Mϕ Increased percentage of Lyve1^+^ and CD206^+^ pro-healing Mϕ	Global *Il-17a* KO	[98,99]
*Il-23p19* KO	Skin wound (6 mm)	C57BL/6	24	**IL23 is required for IL-17 production and its abrogation altered myeloid cells’ infiltration** Reduced IL-17^+^ cells in the granulation Reduced neutrophils and pro-inflammatory Ly6C^+^MHCII^neg^ Mϕ recruitments during the inflammatory phase Increased anti-inflammatory Ly6C^neg^MHCII^+^ Mϕ and percentage of the LYVEl^+^ Mϕ	Global *Il-23p19* KO	[99]
Anti-Vγ4 antibody (Vγ4D)	Skin wound (6 mm)	C57BL/6	6–8	**IL-17A regulates epidermal IL-1β, IL-23 and IGF-1 expression in a dose-dependent manner. Dermal γδ T cells infiltrate the epidermis and supply IL-17A to induce keratinocyte IL-1β production. Keratinocytes IL-1β negatively regulate DETC IGF-1 production.** γδ T cells depletion model Reduced IL-17A protein expression in the wound Improved wound healing by increasing DETC IGF-1 production Reduced IL-1β and IL-23 production in the epidermis + rIL-17A 200 ng/wound (high dose): Eliminated the wound-healing improvement in Vγ4D + rIL-17A on WT wound 200 ng/wound (high dose): Delayed wound closure 2 ng or 20 ng/wound (low or moderate dose): Neither improves nor delays wound closure Decreased DETC IGF-1 production in a dose-dependent manner Increased IL-1β and IL-23 epidermal production in a dose-dependent manner + anti-IL-17A antibody on WT wound 2 ug/wound (low dose): Neither improved nor delayed wound closure 20 ug/wound (moderate dose): Improved wound closure 200 ug/wound (high dose): Defective wound closure Increased DETC IGF-1 production in a dose-dependent manner Attenuated IL-1β and IL-23 in the epidermis in a dose-dependent manner	Neutralising Vγ4 T cells	[100]
*TNF-Rp55* KO	Skin wound (4 mm)	BALB/c	8–12	**TNF has negative effect on skin wound healing. Abrogating TNF signaling improves wound closure** Increased re-epithelization, blood vessel density and collagen deposition. Reduced neutrophils’ and macrophages’ recruitment Reduced E-selectin, *VCAM-1*, *ICAM-1* gene expression Increased gene expression of *IL-1α*, *IL-1β*, *MCO-1*, *MIP-1α*, *MIP-2*, *TGF-β*, *CTGF*, *VEGF*, *VEGFR*	Global *TNF-Rp55* KO	[101]
*K14Cre^ERT^ x Nrf2 fl/fl*	Skin wound (10 mm)	-	6–8	**Nrf2 induces CCL2 production in keratinocytes and promotes Mϕ infiltration. Reciprocally, Mϕ produces EGF to induce keratinocytes’ proliferation.** Knockout model Delayed wound closure Reduce keratinocytes’ proliferation and migration, granulation tissue formation, collagen deposition, and neo-vascularisation Reduced F4/80^+^ Mϕ recruitment and EGF production + rCCL2 on *K14Cre^ERT^ x Nrf2 fl/fl* wound Rescued delayed wound healing Restored Mϕ infiltration Increased keratinocytes’ proliferation	Epidermis-specific *Nrf2* KO	[90]
*Lrig1-EGFP-ires-Cre^ERT^ x Act fl/fl*	Skin wound (5 mm)	C57BL/6	6–8	**Gene knockout of IL-17R complex adaptor *Act1* in the Lrig^+^ HF progenitors reduces Lrig1^+^ cells’ migration and the proliferation of Lrig1^+^ progenies to the IFE.** Delayed re-epithelisation Reduced Ki67^+^ and Np63^+^ keratinocytes	HF progenitor-specific *Act* KO	[102]
*Lrig1-EGFP-ires-Cre^ERT^ x EGFR fl/fl*	Skin wound (5 mm)	C57BL/6	6–8	**IL-17RA interacts with EGFR to promote keratinocytes’ proliferation and migration via p-ERK5.** Delayed wound closure with reduced re-epithelisation Reduced Ki67^+^ and Np63^+^ keratinocytes	HF progenitor-specific *EGFR* KO	[102]
*TcrdCre^ER^ x Rorc fl/fl* and *Rorgt-EGFP*	Splinted skin wound (4 mm)	C57BL/6	7–8	**RORγt^+^** **γδT cells regulate wound epithelial tongue migration via IL-17A/IL-17RC signalling** Control wound Day 3: RORγt^+^ γδT cells expressed highest level of IL-17A and IL-17F Knockout model Delayed re-epithelisation due to impaired epithelial tongue migration Reduced *Hif1a* mRNA and nuclear HIF1α-expressing cells in the Krt14^+^ neo-epidermis. + rmIL-17A on *Rorgt-EGFP* wound Rescued delayed wound closure	Mice lacking γδT cells	[103]
*K14Cre x Il17rc fl/fl*	Splinted skin wound (4 mm)	C57BL/6	7–8	**Gene knockout of *Il17rc* in the Krt14^+^ keratinocytes reduces wound epithelial tongue migration** Delayed re-epithelisation due to reduced epithelial tongue migration No difference in keratinocyte proliferation Reduced nuclear HIF1α-expressing cells in the Krt14^+^ neo-epidermis.	Epidermis-specific *Il17rc* KO	[103]
*K14Cre x Hif1a fl/fl*				**IL-17A/IL17RC signalling is essential for HIF1α induction in the Krt14^+^ keratinocytes. HIF1α regulates keratinocytes’ glycolysis activity and helps with epithelial tongue migration.** Knockout model Impaired epithelial tongue migration +rmIL-17A on the *K14Cre x Hif1a fl/fl* wound Failed to rescue the re-epithelisation defect of the *K14Cre x Hif1a fl/fl*	Epidermis-specific *Hif1a* KO	[103]

**Table 3 biomolecules-12-01659-t003:** Effects of cytokine treatment and cytokine gene knockout on chronic wound-healing outcome.

Model	Wound Type	Strain	Age (Weeks)	Functions/Phenotype	Method	Reference
rCCL2	Skin wound (4 mm)	Balb/c (STZ induced diabetes)	8–10	**rCCL2 treatment rescues impaired wound healing by restoring Mϕ infiltration.** Accelerated wound healing, re-epithelisation, granulation formation Increased blood vessel density, collagen deposition, myofibroblast number Elevated level of VEGF and TGF-β	Chemokine treatment	[55]
*Il-1r1* KO x db/db	Skin wound (5 mm)	C57BLKS/J (db/db mice)	12–14	**IL-1 receptor knockout attenuates IL-1β signaling in diabetic wound.** Promotes wound healing with near complete re-epithelization at Day 9	Global *Il-1r1* KO in obese mice	[119]
IL-1Ra/PIGF topical treatment	Skin wound (5 mm)	C57BLKS/J (db/db mice)	12–14	**IL-1 receptor antagonist delivered with PIGF displays superior wound healing effect compared to IL-1 receptor antagonist alone in diabetic wound.** Improved wound closure Greater angiogenesis, faster clearance of neutrophils and more anti-inflammatory macrophages Reduced fibroblast senescence fibroblasts Reduced pro-inflammatory cytokines and protease Increased anti-inflammatory cytokine, protease inhibitor and growth factor	Recombinant IL-1Ra/PIGF treatment on obese wound	[119]
Anti-IL-1β antibody	Skin wound (8 mm)	C57BL/6 (db/db mice)	12–16	**Neutralising IL-1β promotes a pro-healing microenvironment by downregulating IL-1β activity.** Accelerated wound closure and granulation formation. Increased collagen deposition No difference in blood vessel density Reduced IL-1β, MMP9, TNF and IL-6 Increased IGF-1, TGF-β and IL-10 protein expression	Neutralising antibody treatment	[120]
Anti-IL-17A or anti-IL23 antibody	Skin wound (6 mm)	C57BL/6 (ob/ob mice)	24	**Neutralising IL-17A or -23 improved wound closure rate in obese mice and increased pro-healing Mϕ** Improved wound closure Increased percentage of Lyve1^+^ Mϕ without affecting Tnf and Il-6 expression No difference in scarring	Neutralising antibody treatment	[98,99]
Il-17a KO x db/db	Skin wound (6 mm)	C57BL/6 (ob/ob mice)	24	**Il-17 knockout in obese mice improved wound healing** Improved wound closure Reduced iNOS^+^ cells	Global *Il-17* KO in obese mice	[99]

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
