# Peer review of "Macrophages in Skin Wounds: Functions and Therapeutic Potential"

_biomolecules, 2022, doi:10.3390/biom12111659_

Round 1

Reviewer 1 Report

This is a well-written and comprehensive review that describes the role of macrophages in skin wound healing. That is an interesting topic as chronic wounds represent both a substantial economic burden to healthcare system worldwide and a medical challenge associated with relevant morbidities and mortality. The review discusses the state-of-the-art of skin macrophage subsets and their crosstalk with other major cell types over the four main phases of wound healing. Current gaps of knowledge that are needed to fill are also critically discussed. Overall, I have only a few points that, if they were addressed, the quality of the review would be improved.

1)    Page 7 lines 292-293: I think that this sentence should be rephrased because IL-4 and IL-13 are described as inducers of an alternative (M2) program of polarized activation that, in contrast to the classic M1 activation, lacks or shows a reduced expression of inflammatory genes (cytokines and chemokines), but it is far from being an anti-inflammatory program (Mills, C.D., Kincaid, K., Alt, J.M., Heilman, M.J., and Hill, A.M. (2000). M-1/M-2 macrophages and the Th1/Th2 paradigm. J. Immunol. 164, 6166–6173). IL-10, rather than IL-4 and IL-13, has been associated with an anti-inflammatory M2-like program of macrophage activation (Mantovani A et al. Trends Immunol 2004). Therefore, I think that, rather than striking, it is expected that IL-4 and IL-13 are not the triggers of an anti-inflammatory macrophage reprograming over the course of wound healing.  

2)    Personally, I don’t think that figure 1 describe the different macrophage subsets, as authors affirm (lines 72-73). In contrast, I find that figure tries to show the cross-talk between macrophages and the other stromal cells, keratinocytes and stem cells over the different phases of skin wound healing. In my opinion that figure is not very clear. Therefore, I suggest dividing figure 1 in four figures in order to focus the role of macrophages in each phase of the wound healing process.

Reviewer 2 Report

The manuscript by Sim al. is a review on the role of macrophages in skin wound healing and their possible targeting to resolve pathological healing. The review is wide, precise, and clear and includes two extensive tables helping the reader in cruising the literature. I have few suggestions to improve the work.

1) I suggest to improve description of the role of T cells, including not only gamma/delta T cells but also Treg, conventional Th17 cell, and resident memory CD8+ T cells; and the role played by the costimulatory/coinhibitory axes such as PD-1/PD-1L and ICOS/ICOSL, also capable to modulate macrophage function and recruitment.

2) I suggest to improve Fig.1 by improving legibility of writing and marking the different steps described in the legend with numbers.

Reviewer 3 Report

The review by Sim and colleagues provides a clear and comprehensive overview of the role of macrophages in normal and chronic wounds.  The topic is timely as the role of macrophages in tissue homeostasis and repair is increasingly being recognised.  The review is well structured, detailed, and provides good coverage of the current literature.  The figures and summary tables are also illustrative.  Below are a few minor edits and suggestions for improving the clarity in specific sections.

1.     More detail could be provided on the populations of macrophages identified in humans from single cell studies, including pathways, markers, and differences in healthy vs inflamed skin.  The Reynolds et al 2021 study is cited, but there could be more explanation of the findings.

2.     There are a few typos throughout: Missing spaces in places, line 266 scare should be scar, etc.  Please have a careful proofread before the final submission.

3.     Lines 291-309, the point about state switching and the spectrum of phenotypes is important.  Is it known if individual macrophages in a wound switch their phenotype or is it a replacement of inflammatory cells by anti-inflammatory cells recruited to the wound?  It would be good to either cite the relevant literature or highlight this as an open question.

4.     Some of the writing in 3.3.2 could benefit from additional proof reading for grammatical correctness. There are a several places with article and plural/singular issues.  Also in this section, EndMT could be better defined.  What are the cell behaviours and context for this response.

5.     The first sections of the review do a nice job of describing the different populations of macrophages, but then the discussion of dysfunction in chronic wounds in sections 4 and 5 describes macrophages more generally.  Is it known whether the altered macrophage responses observed in chronic wounds are associated with specific populations or is this an open question in the field?

6.     In line 699, this sentence is not entirely clear.  What is the single cell counterpart?
